# Mitophagy in Cancer: A Tale of Adaptation

**DOI:** 10.3390/cells8050493

**Published:** 2019-05-22

**Authors:** Monica Vara-Perez, Blanca Felipe-Abrio, Patrizia Agostinis

**Affiliations:** 1Laboratory of Cell Death Research and Therapy, Department for Cellular and Molecular Medicine, Campus Gasthuisberg, University of Leuven (KU Leuven), Herestraat 49, B-3000 Leuven, Belgium; monica.varaperez@kuleuven.vib.be (M.V.-P.); blanca.felipeabrio@kuleuven.vib.be (B.F.-A.); 2Laboratory of Cell Death Research and Therapy, VIB-KU Leuven Center for Cancer Biology, 3000 Leuven, Belgium

**Keywords:** mitophagy, mitochondria, autophagy, cancer, tumor microenvironment, anti-cancer therapy resistance, mitochondrial dynamics

## Abstract

In the past years, we have learnt that tumors co-evolve with their microenvironment, and that the active interaction between cancer cells and stromal cells plays a pivotal role in cancer initiation, progression and treatment response. Among the players involved, the pathways regulating mitochondrial functions have been shown to be crucial for both cancer and stromal cells. This is perhaps not surprising, considering that mitochondria in both cancerous and non-cancerous cells are decisive for vital metabolic and bioenergetic functions and to elicit cell death. The central part played by mitochondria also implies the existence of stringent mitochondrial quality control mechanisms, where a specialized autophagy pathway (mitophagy) ensures the selective removal of damaged or dysfunctional mitochondria. Although the molecular underpinnings of mitophagy regulation in mammalian cells remain incomplete, it is becoming clear that mitophagy pathways are intricately linked to the metabolic rewiring of cancer cells to support the high bioenergetic demand of the tumor. In this review, after a brief introduction of the main mitophagy regulators operating in mammalian cells, we discuss emerging cell autonomous roles of mitochondria quality control in cancer onset and progression. We also discuss the relevance of mitophagy in the cellular crosstalk with the tumor microenvironment and in anti-cancer therapy responses.

## 1. Introduction

Mitochondria are double-membrane organelles deputed at cell energy supply; defects in mitochondrial functions not only affect cell homeostasis, bioenergetics and redox control but also are decisive for cell death. In the particular case of cancer cells, mitochondrial-harbored metabolic pathways are rewired to meet the increased bioenergetics and biosynthetic needs of the cancer cells and to handle oxidative stress. Therefore, a tight control of the mitochondrial network homeostasis is essential for cancer cells.

Several highly interrelated mechanisms, including mitochondrial dynamics (fusion and fission) as well as macroautophagy (mitophagy), operate in mammalian cells as key mitochondrial quality control processes, and their implication in tumor development and progression has recently been elucidated. In particular, the selective removal of mitochondria through the process of mitophagy has been recently implicated in reshaping the metabolic landscape within cancer cells and the interaction between cancer cells and other key components of the tumor microenvironment (TME), to foster the adaptive and survival ability of cancer cells. Moreover, and considering the tight relationship between mitochondrial homeostasis and susceptibility to cell death, mitochondria quality control and mitophagy in primis are critical in anti-cancer therapeutic response as well as cancer-related off target effects. 

In this review, after a brief introduction of the main mitophagy pathways, we discuss the interplay of mitophagy with the key pathways involved in tumorigenesis, its coordination of the TME and its implication in the success (or not) of current anti-cancer therapies.

## 2. Molecular Mechanisms Leading to Mitophagy

Macroautophagy (hereafter referred to as autophagy) is a self-degradation process which is typically stimulated under conditions of nutrient deprivation or cellular stress. During autophagy, proteins, macromolecules and/or organelles are engulfed in a double-membrane vesicle called autophagosome, which eventually fuses with the lysosome where cargo degradation takes place (for recent reviews on mechanisms of autophagy, see [1,2]). The breakdown of intracellular material allows the recycling of essential building blocks to occur for metabolic and biosynthetic pathways. In mammalian cells, ubiquitylation operates as a prominent—albeit not unique—mechanism to selectively tag cytoplasmic cargoes destined for degradation by the autophagic machinery. Ubiquitylated targets are then recognized by specific autophagy receptors (such as p62/SQSTM1 and optineurin (OPTN); for a review on the topic, please see [3]) which are capable of binding both ubiquitin and the lipidated members of the ATG8 family of pro-autophagic proteins (LC3A/LC3B/LC3C/GABARAP/GABARAPL1/GABARAPL2, reviewed in [4]) via their LC3-interacting domain (LIR). 

Mitophagy is a specialized form of autophagy in which damaged, dysfunctional or obsolete mitochondria are recognized by the autophagy machinery and eventually degraded by the lysosome. Damaged mitochondria are, in general, those mitochondria which are not able to execute oxidative phosphorylation (OXPHOS) efficiently. This is mainly because of the dissipation of their transmembrane potential and consequent accumulation of reactive oxygen species (ROS) causing an increase in the overall cellular oxidative stress levels, precipitating mitochondria-mediated cell death [5]. Since mitochondria are not found as isolated organelles but as a highly dynamic network, the dysfunctional mitochondrion needs to be separated from the healthy network, requiring the tight coordination between fusion, fission and mitophagy machineries (see Box 1 for a summary of the fusion and fission mechanisms). In particular, depolarized mitochondria will be either not able to fuse with the healthy mitochondrial network or isolated from the network by fission, resulting in isolated mitochondria ready to be degraded by mitophagy (for extensive reviews on the topic, see [6,7]). Instead, elongated mitochondria are spared from degradation and remain bioenergetically functional [8,9]. Isolated and damaged mitochondria are then recognized by specific mitophagy receptors whose identity depends on the specific trigger causing mitochondrial clearance, and which function as molecular bridges for the interaction with the autophagy machinery [10].

Box 1Mitochondrial dynamics.
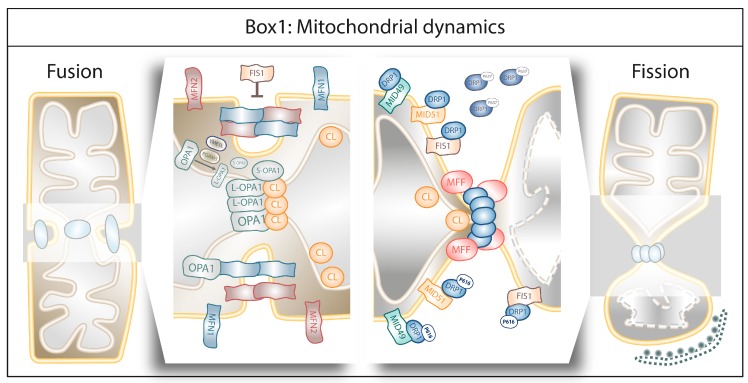
Fusion is the union of two separate mitochondria into a single entity to guarantee at least one copy of mitochondrial DNA (mtDNA) per mitochondrion as well as efficient OXPHOS [11]. Fusion is a highly coordinated process chiefly controlled by the mitofusins 1 and 2 (MFN1 and MFN2) anchored to the outer mitochondrial membrane (OMM) which, by forming homo- or heterodimers, coordinate fusion of the two OMMs, whereas Opa1 and cardiolipin (CL) orchestrate the fusion of the inner mitochondrial membranes (IMMs) [11]. MFN1 and MFN2 are required in both OMMs to assist membrane fusion in contrast to Opa1, whose presence in IMM is sufficient to enable the fusion [12]. Opa1 interacts with MFN1 and disruption of this interaction results in mitochondrial fragmentation [13]. Moreover, Opa1 cleavage by the mitochondrial metalloproteases OMA1 and YME1L results in one long isoform (L-Opa1) that remains anchored to the IMM and one short isoform (S-Opa1) that is released in the inter-membrane space (IMS) [14,15]. Although increased Opa1 cleavage is often associated with mitochondrial dysfunction [16], Opa1 proteolytic activation is required for mitochondrial fusion [17] since L-Opa1 is essential for IMM tethering [18] and S-Opa1 enhances the interaction of L-Opa1 and CL during IMM fusion [19].In contrast, fission refers to the excision of one part of the mitochondrion from the main organelle. This separation can occur either symmetrically, where the two resulting mitochondria have similar respiratory capacity and membrane potential, or asymmetrically, where fission will result in one healthy mitochondrion and one damaged, the latter being targeted for degradation by mitophagy [20]. Fission is coordinated by the dynamin-related protein 1 (DRP1). DRP1 is a cytosolic protein that will translocate to the OMM and induces mitochondrial excision by interacting with OMM-anchored receptors, including the mitochondrial dynamics proteins of 49 and 51 kDa (MiD51, MiD49) [21]. Once at the OMM, DRP1 self-assembles to form a constricting ring around the portion of the mitochondrion to be excised [22]. The self-assembly of the different DRP1 isoforms into the constriction ring is assisted by other fission elements like Mff [23] and CL [24]. The role of FIS1 in mitochondrial fission is controversial, since it was initially described an OMM-anchored receptor assisting the mitochondrial localization of DRP1 [25] but recent data showed that FIS1 can induce mitochondrial fragmentation by inhibiting the GTP-ase activity of the fusion machinery (MFN1, MFN2 and Opa1) [26]. Phosphorylation of different serine residues within DRP1’s GTP-ase domain will enhance (serine 616) or diminish (serine 637) its affinity for the OMM and its receptor molecules [27]. The division site for the recruitment of DRP1 at the mitochondria is marked by the endoplasmic reticulum (ER). The ER wraps mitochondria and enables mitochondria constriction at the ER-mitochondria contact sites [28,29]. A pivotal role is played by the ER-bound protein inverted formin 2 (INF2) which by controlling actin assembly induces constrictions, before DRP1 recruitment to the mitochondria ([29]; for extensive reviews on this subject, see [30,31]).

Below we provide an overview of some of the best characterized (canonical) or emerging mitophagy pathways described to date in mammalian cells. 

### 2.1. Canonical Mitophagy Pathways

#### 2.1.1. PINK1/Parkin-Mediated Mitophagy

The PINK1/Parkin pathway is the most studied pathway of mitophagy (Figure 1) [32,33]. The serine/threonine PTEN-induced putative kinase 1 (PINK1) is the initiator of this pathway. Under normal conditions, PINK1 is imported to the IMM via the Translocase of the Outer Membrane (TOM) and Translocase of the Inner Membrane (TIM) complexes, where PINK1 is cleaved by the presenilin-associated rhomboid-like (PARL), a mitochondrial resident rhomboid serine protease, and subsequently degraded by the (mitochondrial) proteasome, keeping PINK1 levels low under basal conditions [34,35]. The translocation of PINK1 towards the IMM is membrane-voltage-dependent; therefore, mitochondrial depolarization abolishes PINK1 translocation [34]. Moreover, and in response to mitochondrial depolarization, PDK2 phosphorylates PARL, rendering it unable to cleave PINK1 [34,36]. Both events result in PINK1 accumulation at the OMM, where it recruits the U3 ubiquitin ligase Parkin [37]. PINK1 phosphorylates serine 65 of Parkin’s ubiquitin-like domain, promoting the E3 ubiquitin ligase activity of Parkin. Once activated, Parkin will ubiquitinate key mitochondrial proteins (such as MFN1, MFN2, VDAC1 or Miro1), a process that contributes to the isolation of the damaged mitochondria. On the one hand, Parkin mediated ubiquitylation of MFN1 and MFN2 prevents them to engage in fusion [38,39]. On the other hand, upon ubiquitylation Miro1, a protein that attaches mitochondria to the microtubules, will no longer bind the microtubules but the PINK1-Parkin complex, supporting the isolation the damaged mitochondrion [40,41]. Furthermore, these ubiquitin chains are phosphorylated by PINK1, which triggers further cycles of Parkin recruitment and activation, leading to the amplification of the mitophagy signal [42]. Finally, polyubiquitylation of mitochondrial proteins will be recognized by the autophagy cargo adaptors p62 and OPTN [43] and interaction with LC3, forming a complex that is degraded by the autophagic machinery [44]. 

#### 2.1.2. BNIP3/NIX-Mediated Mitophagy

BCL2/adenovirus E1B 19 kDa protein-interacting protein 3 (BNIP3) and BNIP3-like (BNIP3L/NIX) are LIR-containing proteins located at the OMM which are able to directly bind LC3/GABARAP and trigger mitophagy without requiring protein ubiquitination or additional adaptors (Figure 1) [45,46,47]. They belong to the BH3-only group of the BCL-2 family of proteins, although their ability to bind BCL-2 remains context dependent [48]. Both BNIP3 and NIX are under the transcriptional regulation of Hypoxia-Inducible Factor 1 alpha (HIF1α), and are therefore thought to mediate hypoxia-triggered mitophagy [49,50]. However, in particular conditions, BNIP3 and NIX are also under the transcriptional regulation of FOXOa3 [51,52] or NF-kB [53], suggesting their participation in signaling pathways beyond hypoxia. Regarding their interactors, BNIP3 and NIX have been shown to form homodimers and heterodimers [54,55], as well as to separately interact with Mieap (a p53-regulated mitochondrial protein associated with the removal of oxidized species [56]) and cadherin6 (CDH6, protein modulator of mitophagy and DRP1-mediated fission [57]) to guarantee mitochondrial homeostasis. Moreover, Opa1 has been shown to interact with BNIP3 to promote mitochondrial fragmentation [58] and phosphorylation of BNIP3 in its C-terminal domain disrupts Opa1-BNIP3 interaction promoting mitophagy and reducing cell death [59]. Phosphorylation of BNIP3 and NIX within their LIR domain (serines 17 and 24 for BNIP3 and serines 34 and 35 for NIX) increases their affinity for LC3 binding [60,61], suggesting that phosphorylation is the main molecular switch controlling their pro-mitophagy or pro-death activity.

#### 2.1.3. FUNDC1-Mediated Mitophagy

FUN14 Domain Containing 1 (FUNDC1) is another OMM protein capable to bind directly LC3 and trigger mitophagy (Figure 1) [62,63]. Alike BNIP3 and NIX, also FUNDC1 can trigger hypoxia-induced mitophagy [62]—even though it is not a HIF1α target [64]—and its phosphorylation in the LIR domain (serine 17) can increase its affinity for LC3 [65]. Interestingly, FUNDC1 can interact with both Opa1 and DRP1 and this interaction is modulated by FUNDC1 phosphorylation on serine 13. Dephosphorylation of FUNDC1 by phosphoglycerate mutase 5 (PGAM5), a mitochondrial serine/threonine protein phosphatase activated by PARL cleavage under condition of loss of mitochondria membrane potential [66], reduces Opa1-pFUNDC1 complex, which in turn increases FUNDC1′s affinity for DRP1 and promotes mitochondrial fragmentation and mitophagy [67,68]. Alternatively, the E3 ubiquitin ligase MARCH5 ubiquitylates FUNDC1 and DRP1 to fine-tune the mitophagic upregulation in response to hypoxia [69]. 

### 2.2. Non-Canonical Mitophagy Pathways

#### 2.2.1. Lipid-Mediated Mitophagy

CL is a phospholipid which is particularly enriched in the IMM, which is able to directly interact with GABARAP upon translocation from the IMM to the OMM [70,71] in response to loss of mitochondrial transmembrane potential mediated by the hexameric intermembrane space protein NDPK-D [72]. Prohibitin 2 (PHB2) is an IMM mitophagy receptor which can bind LC3 upon membrane depolarization and proteasomal rupture favoring Parkin-mediated mitophagy [73]. Mechanisms of mitophagy induced by PHB2 have been associated with CL redistribution within the mitochondrial membranes [74], although PHB2′s contribution to CL-mediated mitophagy has not yet been elucidated. Additionally, ceramides -sphingolipids present in the OMM- can bind LC3B-II to target the mitochondria for degradation upon DRP1-mediated mitochondrial fragmentation [75]. Notably, ceramide 18 (Cer18) binding to LC3B-II inhibits mitochondrial function and oxygen consumption, induces autophagic cell death in vitro and suppress tumor growth in vivo [75], suggesting that lipid binding to LC3 may regulate the functional outcome of mitophagy.

#### 2.2.2. AMBRA1-Mediated Mitophagy

Another mitophagy receptor is Autophagy And Beclin 1 Regulator 1 (AMBRA1), a Parkin interactor [76] that directly binds LC3 and triggers mitophagy [77] to protect the cells from apoptotic cell death caused by oxidative stress [78]. Mechanistically, IKKα phosphorylates AMBRA1 at serine 1014, enhancing its affinity for the ATG8 members GABARAP/GABARAPL1/GABARAPL2 and IKKα inhibition impairs AMBRA1-mediated mitophagy [79]. Furthermore, after membrane depolarization AMBRA1 promotes the mitochondrial localization of the E3 ligase HUWE1, which by ubiquitylating MFN2 promotes mitophagy [79]. Of note, HUWE1 absence also abolishes AMBRA1-mediated mitophagy, suggesting that HUWE1 could also modulate AMBRA1 serine 1014 phosphorylation [79], although the precise mechanism has not yet been elucidated. Overexpression of AMBRA1^ActA^ (a fusion protein specifically expressed at the mitochondria) exacerbates mitophagy and suppresses oxidative stress and apoptosis induced by mitochondrial poisons [78], indicating the cytoprotective role of this pro-mitophagic protein.

#### 2.2.3. BCL2L13-Mediated Mitophagy

The mammalian ortholog of the only mitophagy receptor in yeast (atg32) is BCL2L13/BCL-RAMBO, an OMM protein able to bind LC3 and able to induce DRP1-independent mitochondrial fragmentation [80]. Not only BCL2L13 induces Parkin-independent mitophagy [80] and has been associated with NIX/FUNDC1-mediated mitophagy [81], but it is also involved in autophagy-independent mitochondrial quality control [82]. 

#### 2.2.4. FKBP8-Mediated Mitophagy

The recently described mitophagy receptor FK506-binding protein 8 (FKBP8/FKBP38), a known inhibitor of the mammalian target of rapamycin (mTOR), is a protein anchored to the OMM which is able to bind lipidated LC3A/GABARAP; its overexpression promotes mitochondrial fission in a similar fashion as BNIP3 or NIX overexpression [83]. However, FKBP38 operates as a preferential LC3A recruiter, suggesting that depending on the expression of LC3 members, a different panel of mitophagy receptors is engaged during mitophagy. Strikingly, FKBP8 is not degraded in the autophagosome during mitophagy, but once the mitochondria have been targeted for degradation, FKBP8 escapes from degradation by relocating to the ER, where it exerts an antiapoptotic effect due to its ability to bind Bcl2 [83,84]. 

#### 2.2.5. Rab-Mediated Mitophagy

Endosomal trafficking and recycling governed by the Rab family of small GTPases contributes to the mitophagic removal of mitochondria. Recent studies have highlighted mechanisms for the removal of mitochondria that are dependent on the activity of several Rab-GTPases. A recently described pathway, independent of the classical ATG5/ATG7-LC3A/B/C autophagosome axis but regulated by the Beclin1 and ULK1 complex, involves a Rab9-mediated vesicular mechanism where autophagosomes are generated by the fusion of isolation membrane with vesicles derived from the trans-Golgi and late endosomes [85]. Recent work shows that another regulator of Rab activity, RABGEF1, a guanine nucleotide exchange factor (GEF) of endosomal proteins, is recruited in a Parkin-mediated fashion to damaged mitochondria. Mitochondria associated RABGEF1 enables the targeting of Rab5 and Rab7a to the damaged mitochondria and promotes ATG9-mediated vesicle assembly and the subsequent autophagosomal encapsulation [86].

Mitochondria can also be sequestered into Rab5-decorated endosomes through the endosomal sorting complex (ESCRT) and delivered to the lysosomes for degradation [87]. For this pathway, Parkin-mediated ubiquitylation of mitochondria is necessary while the canonical autophagy machinery is dispensable. The reason why mitochondria would be cleared preferentially by the endocytic pathway instead of using the autophagy machinery is not completely clear, even if it is apparent that redundant pathways of mitochondria degradation exist in mammalian cells. A possibility is that endosomes act as a first line of defense, before autophagy is stimulated, to rapidly remove potentially damaging dysfunctional mitochondria [87].

## 3. Mitophagy and Cancer 

Most of the above described proteins involved in the mitophagic processes have been shown to be dysregulated in cancer patients (Table 1), but whether they behave as tumor promoter or tumor suppressor seems to be highly dependent on the cancer subtype and context [32,88]. For example, the pro-mitophagic receptor BNIP3 has tumor suppressor functions in breast cancer (Table 1), whereas it is thought to have a tumor promoter role in melanoma, renal cell carcinoma and pancreatic cancer (Table 1). To discuss how mitophagy contributes to tumor progression, we will first review the interplay between mitophagy with the key pathways involved in tumorigenesis in terms of metabolic modulation. Later, we will discuss their impact on the modulation of the TME. 

### 3.1. Mitophagy Modulators and Cancer Metabolism

Metabolic reprogramming is one of the hallmarks of cancer [199]. Cancer cells need to plastically rewire their metabolism to fulfill the three basic needs of dividing cells: rapid ATP generation to maintain energy status, metabolic precursor supply to meet the high rates of macromolecule biosynthesis and maintenance of an appropriate cellular redox status [200]. To do so, cancer cells have acquired the ability to use a variety of fuel sources to adapt their metabolism according to their needs and to cope with metabolic and nutrient stresses. Growing evidence indicates that autophagy supports the metabolic plasticity of cancer cells, by providing virtually all essential components of carbon metabolism through the degradation of carbohydrates, proteins, lipids and nucleotides (recently reviewed in [201]). Several genetic studies support the current view that both glucose-dependent metabolic pathways and mitochondria metabolism are pivotal in tumorigenesis [199,200,202,203].

In the following subsections, we will discuss emerging links connecting oncogene-driven metabolic pathways and key modulators of the mitophagic machinery.

#### 3.1.1. Mitophagy and Its Contribution to the Warburg Effect

A main metabolic phenotype observed in cancer cells is driven by the Warburg effect, which consists in the shift from ATP generation through OXPHOS to ATP generation through glycolysis, even in the presence of oxygen [204], being the transcription factor HIF-1α one of its major drivers. Stabilization of HIF-1α subunit under hypoxia activates the expression of the glycolytic program by encoding glucose transporters and glycolytic enzymes, as well as by promoting the conversion of pyruvate into lactate instead of its incorporation in the tricarboxylic acid (TCA) cycle [205]. Notably, HIF-1α transcriptional program also comprises the pro-mitophagic receptors BNIP3 and NIX, which, by instigating mitophagy, would diminish the mitochondrial mass thereby reducing the overall oxygen consumption of the cell and promoting its survival under low oxygen conditions [49,205] (Figure 2a). Cancer cells have also developed O_2_-independent mechanisms to stabilize HIF-1α under normoxia and drive tumorigenesis, highlighting the essential pro-glycolytic role of this transcription factor. Moreover, changes in the expression levels of the mitophagy receptors BNIP3 and NIX can feedback on HIF-1α stability. For instance, loss of BNIP3 in a mouse model of mammary tumorigenesis reduces mitophagy and increases mitochondrial ROS levels, which results into increased normoxic HIF-1α stabilization, eventually promoting the Warburg effect and subsequently tumor progression [206] (Figure 2b). A similar effect has also been observed in the human breast cancer cell line MCF-7 in vitro, where insulin-like growth factor 1 (IGF-1) induces BNIP3 expression in a HIF-1α dependent-manner. However, MCF-7 cells with acquired resistance to an IGF-1 receptor kinase inhibitor show reduced BNIP3 levels, impaired mitophagy, accumulation of dysfunctional mitochondria and increased ROS production, leading to increased ATP production through glycolysis [207]. Likewise, in glioblastoma cells, loss of PINK1 promotes the Warburg effect by the ROS-dependent stabilization of HIF-1α and reduced pyruvate kinase muscle isozyme 2 (PKM2) activity, both key regulators of aerobic glycolysis [208]. 

Glycolysis is also modulated by the interplay between the tumor suppressor p53 and mitophagy. A p53-BNIP3 axis modulates the glycolytic flux in radioresistant head and neck squamous cell carcinoma cell lines [209]. In these radioresistant cancer cells, BNIP3-dependent clearance of abnormal mitochondria reduces the glycolytic shift while maintaining oxygen consumption only in the presence of p53 (Figure 2c). Although the molecular mechanism linking p53 to BNIP3 is still unclear, BNIP3 is the dominant mitophagy receptor since loss of Parkin, is ineffective [209]. This is interesting, since Parkin is a p53-regulated gene mediating the effects of p53 on mitochondria energy metabolism, antioxidant defense and irradiation-induced tumorigenesis [210,211]. Hence, while these studies suggest that mitophagy contributes to p53-mediated effects on cancer metabolism and tumorigenesis they also highlight that, depending on specific cancer context, the nature of the mitophagy receptor involved could be critical. Additionally, other mitophagy-unrelated functions of BNIP3 [212] and Parkin [210] could play a role, depending on the cancer and type of stress considered. 

High glycolytic rates in cancer cells are controlled by other key oncogenes, such as c-Myc and K-RAS. Besides coordinating with HIF-1α the expression of several glucose transporters and glycolytic enzymes [213,214], c-Myc also modulates mitophagy by regulating choline metabolism. In B-lymphoma cells, c-Myc activates the transcription of the key enzyme phosphate cytidylyltransferase 1 choline-α (PCYT1A) and PCYT1A upregulation prevents lymphoma cells to undergo a mitophagy-dependent necroptosis [215]. 

#### 3.1.2. Mitophagy and OXPHOS

As mentioned, while cancer cells engage in aerobic glycolysis, and some tumors rely mostly on this pathway to meet their bioenergetic demands, they also strive to maintain pools of respiring mitochondria to adjust their metabolic and biosynthetic requirements [216]. This cancer cell autonomous plasticity requires that pathways controlling mitochondria clearance and biogenesis are intricately linked. Clear examples are the key regulator of mitochondrial biogenesis, peroxisome proliferator-activated receptor gamma coactivator-1 alpha (PGC-1α) and c-Myc. The transcriptional coactivator PGC-1α is downregulated by HIF-1α to support the glycolytic switch in low oxygen condition (reviewed in [217]). Oncogenic c-Myc coordinates a vast array of genes involved in cell cycle control and glycolysis, but it is also a key promoter of mitochondrial biogenesis and overall mitochondria metabolism. Mitochondrial biogenesis strengthens c-Myc’s effects on cell-cycle progression and glycolytic metabolism, enabling cancer cells with the metabolic flexibility that supports growth (Figure 2d). Another important pathway involved in the clearance of damaged mitochondria is the MAPK pathway, which stabilizes PINK1 and subsequently promotes mitophagy in response to ROS-induced stress [218]. 

Inefficient OXPHOS caused by a leaky or defective electron transport chain can lead to ROS production [219,220]. Interestingly, high OXPHOS activity induced by feeding HeLa cells with glutamine is also coupled to enhanced mitophagy, through a mechanism involving the translocation of the small GTPase Ras homolog enriched in brain protein (Rheb) to the mitochondria and its binding to NIX. Although Rheb can interact with both BNIP3 and NIX, resulting in the blockade of Rheb-mediated activation of mTORC1, mitophagy coupled to increased OXPHOS is both mTORC1- and BNIP3-independent [221]. Thus, although mainly operating in concert, BNIP3 and NIX may affect mitochondrial degradation through independent mechanisms or in a cancer-subtype specific fashion. In line with this, silencing of BNIP3 in melanoma cells blunts glutamine-mediated effects on melanoma cell growth, migration and bioenergetics [222], suggesting that BNIP3 is vital to maintain mitochondria fitness required for glutamine-driven melanoma aggressiveness.

Hence, mitophagy may prevent the accumulation of damage that is inherently associated with elevated mitochondrial metabolism, thereby maintaining the (re)generation of mitochondria that are metabolically adapted to cope with the metabolic and nutrient stress from the TME. An interesting conjecture proposes that the coordinated induction of mitochondria biogenesis and mitophagy may be used to generate pools of mitochondria that are better suited to catabolize fatty acids through fatty acid oxidation (FAO) [33]. Given that RAS-driven tumors require elevated autophagy to preserve mitochondrial function and proficient FAO ([223] and reviewed in [201]), this conjecture seems indeed plausible (Figure 2d). This is particularly important, considering that FAO is emerging as crucial fuel for aggressive cancer types, like breast cancer cells [224], and that acetyl-CoA from oxidized fatty acids is a key regulator of epigenetic remodeling of chromatin [225], which may further support metabolic rewiring in cancer cells. In this scenario, PGC1α could be a key player, as specifically regulates FAO [226]. Actually, upregulation of PGC1α in nasopharyngeal carcinoma provides resistance to radiation by promoting FAO [227]. In addition, PGC1α regulates mitophagy during myogenesis by buffering ROS production, which can cause mitophagy at high levels [228]. Considering this, it could be possible that PGC1α contributes to the aggressiveness of tumors by connecting lipid metabolism, mitophagy and mitochondrial homeostasis.

The role of mitophagy, as for autophagy more in general, might be also modulated during the various phases of tumorigenesis [229]. Notably, AMBRA1 may also influence cancer metabolism and tumor progression by regulating the degradation of c-Myc. Mechanistically, AMBRA1 favors the interaction between c-Myc and its phosphatase PP2A, which leads to the dephosphorylation and degradation of c-Myc. This interaction is enhanced when mTOR is inhibited, reducing the cell division rate [230]. In addition, HUWE1, the E3 ubiquitin ligase involved in AMBRA1-mediated mitophagy, also participates in c-Myc degradation, suppressing RAS-driven tumorigenesis by preventing c-Myc/Miz1 accumulation [231]. AMBRA1 mutant mice develop spontaneous tumors [232], suggesting a tumor suppressor role for AMBRA1. However, considering that besides regulating c-Myc degradation and mitophagy, AMBRA1 plays additional functions in cancer cells [78], it remains challenging to pinpoint the exact contribution of AMBRA1-mediated mitophagy in tumorigenesis. In contrast, previous studies have shown that autophagy favors RAS-mediated transformation by supporting glycolysis [233] and mitochondria metabolism [234]. Additionally, loss of autophagy in K-RAS [223] or in mutant B-RAF-driven [235] lung cancer models impairs tumor growth and switches tumor fate from carcinomas to benign tumors, called oncocytomas. Interestingly, in both cases this inhibition of tumorigenesis is associated with the accumulation of defective mitochondria, suggesting mitophagy impairment.

Transformation mediated by c-Myc or RAS-v12 overexpression increases AMP-activated protein kinase (AMPK) phosphorylation, favoring the activation of FoxO3, which upregulates the expression of genes such as BNIP3 and LC3 [236]. In addition, Hepatitis B virus X protein (HBx), a leading factor in Hepatitis B virus-related hepatocellular carcinoma, promotes PINK1-Parkin mediated mitophagy through the activation of the mitochondrial peptidase LON under starvation, which could be a determinant event in the development of hepatocellular carcinoma [237]. 

Altogether, these studies reveal that mitophagy either directly or indirectly impacts the metabolism of cancer cell, but if tumor progression is favored by mitophagy or not, will depend on the mitophagy pathway involved, type of tumor and possibly on the stage of tumor development. 

Finally, it should be considered that beyond the role of mitophagy in cancer cells, stromal cells’ autophagy and perhaps mitophagy (as discussed further below) are emerging as key contributors to tumor progression, by providing essential amino acid fueling metabolism in cancer cells [238,239]. 

#### 3.1.3. Mitophagy and Iron Metabolism

Iron homeostasis is essential for numerous cellular processes, and either too much or too little iron can be detrimental for cell survival. Iron is fundamental for cell growth, but excessive iron accumulation induces the production of ROS and oxidative injury [240]. Iron trafficking is regulated at both the systemic and organellar level, being mitochondrial functions crucial in maintaining cellular iron homeostasis. Iron is transported to the mitochondrion for the biosynthesis of heme and iron-sulfur clusters through mitochondrial iron importers such as SLC25A37/mitoferrin-1 and SLC25A28/mitoferrin-2 [241,242]. Dysfunction of mitochondrial iron trafficking plays an important role in mitochondrial diseases as well as cancer. Recently, it has been described that PINK1 and Parkin regulate mitochondrial iron accumulation in pancreatic cancer [127,243]. Depletion of PINK1 and Parkin in mice accelerates K-RAS-driven pancreatic tumorigenesis due to mitochondrial iron accumulation. Since PINK1 and Parkin mediate autophagic degradation of SLC25A37 and SLC25A28, PINK1- or Parkin--deficient mice show increased SLC25A37 and SLC25A28 levels, leading to mitochondrial iron accumulation, activation of the HIF1α glycolytic program, ultimately promoting the Warburg effect. This effect was rescued by genetic depletion of HIF1α or by deferiprone treatment, a mitochondrial iron chelator. Notably, K-RAS-driven pancreatic tumorigenesis was also inhibited, suggesting that HIF-1α metabolic reprogramming induced by mitochondrial iron accumulation contributed to pancreatic tumorigenesis in *Pink1*- or *Park2*-deficient mice. Furthermore, in a STAT3 deficient model of colorectal cancer, elevated mitophagy in intestinal epithelial cells (IECs) caused an accumulation of iron (II) in lysosomes, provoking lysosomal membrane permeabilization. This enabled antigen processing and stimulation of dendritic cells-mediated CD8+ T cells-induced anti-tumor immunity [244]. 

Hence, these studies support the view that mitophagy or mitophagy players take part in the control of iron trafficking at organellar level in cancer cells and stromal cells, affecting tumor progression at different molecular and cellular levels; from metabolic reprogramming to anti-tumor immunity regulation. Considering the emerging role played by mitochondria during iron-dependent ferroptosis—a regulated form of necrosis to which drug-resistant cancer cells are particularly vulnerable [245]—it is tempting to assume that mitophagic-control of iron metabolism in cancer cells may become a druggable target in cancer therapy.

### 3.2. Mitophagy and Cancer Stem Cells 

Tumors are complex cellular systems where different subpopulations of cells coexist. Cancer stem cells (CSCs) constitute one of these subpopulations, which is characterized by their ability of self-renewal, dedifferentiation, generating the bulk tumor cells, and metastatic potential. CSCs are not a fixed entity and phenomena of dedifferentiation of mature tumor cells to CSCs can occur [246]. Therefore, the interaction of CSCs and tumor cells with the TME is important to decide cancer cell fate. Mitophagy also plays a role in the regulation of CSC subpopulation, since it is not only involved in the promotion of the stemness, but also in the acquisition of chemoresistance. Mitophagy regulates hepatic CSC subpopulation by suppressing p53 activity [247]. Increased Parkin-mediated mitophagy has been shown to promote p53 co-localization with mitochondria, resulting into simultaneous p53 and mitochondria degradation in a mitophagy-dependent manner [247]. Mitophagy inhibition leads to PINK-mediated p53 phosphorylation at serine 392, provoking p53 translocation into the nucleus and binding to the NANOG promoter, which prevents the activation of NANOG expression by the OCT4 and SOX2 transcription factors [247]. Since NANOG is an essential transcription factor for maintaining the stemness of CSCs [248], this mechanism would reduce the hepatic CSC subpopulation. Exacerbated mitophagy also promotes stemness in esophageal squamous cell carcinoma cells, as Parkin-dependent mitophagy was found to increase the expression of the stem cell marker CD44 in cancer cells undergoing epithelial–mesenchymal transition (EMT) [249]. Metabolic regulation is also a key determinant of the stem phenotype in cancer cells and mitophagy could play an important role in this context too. In line with this, in lung cancer and nasopharyngeal carcinoma, CSCs exhibit reduced mitochondrial mass compared to non-CSCs [250,251]. 

Hence, it seems that the role of mitophagy in CSCs may largely depend on the affected regulators and signaling pathways that control the differentiation of mature tumor cells to CSCs or the maintenance of stemness.

### 3.3. Non-Autonomous Effects of Mitophagy: Mitochondrial Transfer

Although mitophagy is crucial to preserve cell homeostasis, under certain conditions, either healthy or damaged mitochondria can be exchanged between cells to improve the OXPHOS capacity of the receiving cells or be degraded by the mitophagy machinery of the receiving cell, in a process called transmitophagy [252,253]. For instance, cancer associated fibroblasts (CAFs) in contact with breast cancer cells have been shown to have a radically different mitochondrial network than that of non-cancerous fibroblasts, suggesting that a mitochondrial exchange is orchestrated by the cancer cells [254]. In this scenario, CAFs are the highly-mitophagic donor cells and the cancer cells the recipients wanting to maintain their high OXPHOS status. In another cancer model, B16 murine melanoma cells depleted from mitochondria (ρ0) and injected in syngeneic mice have been shown to incorporate mitochondria from the host to maintain their “OXPHOS addiction” as well as tumor growth [255]. Similarly, mitochondrial transfer between cells within astrocytomas promotes tumor growth [256]. Additionally, leukemic cells have been shown to accept intact mitochondria from stromal cells to increase their OXPHOS capacity and resist the loss of membrane potential induced by different chemotherapeutic treatments [257]. In fact, there is a general consensus that stromal cells, mainly mesenchymal stem cells (MSCs) or fibroblasts, are the source of healthy mitochondria for cancer cells [253,258,259,260], although endothelial cells [261,262] or macrophages [257] have also been reported as stromal cells involved in the mitochondrial exchange.

Although several studies suggest that mitochondrial transfer is an active process, the mechanistic underpinnings of this process remain highly controversial. Mitochondrial transfer could require direct cell-cell contact between the donor and acceptor cell [257], mainly in the form of nanotubes. Nanotubes are F-actin based membrane tubes that can transport mitochondria, among other intercellular structures, to relieve mitochondrial stress in the acceptor cell [263] which have been observed within tumoral structures [264]. Most of the mitochondrial transfers using nanotubules in a cancer context mainly refer to either MSCs as donor cells and a cancer cell as acceptor [265,266] or cancer cell donor to cancer cell acceptor [264,267]. On the other hand, other studies suggest that contact between donor and acceptor is not necessary, as (fragments of) mitochondria could be trafficked and released via extracellular vesicles [253,258,259]. For example, pro-inflammatory myeloid-derived regulatory cells secrete extracellular vesicles containing depolarized mitochondria, which can be incorporated by T-cells [268]. T-cells and prostate cancer cells-derived extracellular vesicles have also been shown to contain mitochondrial fragments marked by the presence of various mitophagy regulators (e.g., PINK1, MFN1, MFN2…) and mtDNA [91,269]. It has also been claimed that the functionality of the mitochondria is irrelevant for their transfer, while intact mtDNA is indispensable for mitochondria to be transferred to another cell [270]. Furthermore, mtDNA transfer alone (not the complete mitochondrion) from the host to ρ0 cancer cells was shown to be sufficient to recover the respiratory capacity of mitochondria-deficient cancer cells [271]; however, the transfer mechanisms remain elusive. Interestingly, few reports suggest that extracellular vesicles could also travel within nanotubes and thus, these two mechanisms are perhaps not mutually exclusive [272,273]). What is clear is that both mechanisms are highly dependent on cytoskeletal dynamics and in line with this, cytoskeletal disruption by vincristine [257] or cytochalasin D [270] treatment hampers mitochondrial transfer. In this regard, Miro1 overexpression in MSCs favor the donation of healthy mitochondria to relief cellular stress in the acceptor cell [259,274,275] 

Together, these studies highlight that intercellular mitochondrial transfer is an emerging mechanism of mitochondrial quality control as well as a crucial mechanism of interaction between cancer and stromal cells. However, the mechanistic underpinnings of this process are still largely elusive and more studies are needed to fully appreciate how and when mitochondrial transfer between cancer cells and their stroma contributes to cancer progression.

### 3.4. Mitophagy, Innate Immunity and Inflammation

Mitochondria are emerging as key modulators of cellular danger signaling and systemic immunity responses aiming to restore cellular or tissue homeostasis. This is perhaps not surprising since mitochondria are ancestral remnants of a-proteobacteria (endosymbiotic theory) and, as such, the mitochondrial genome (mtDNA) harbors CpG DNA repeats and encodes for formylated peptides (for a recent review on the topic, see [276]. Upon cellular injury or death, various mitochondrial factors or products (including but not limited to mtDNA, CL, ROS, ATP, cytochrome c or N-formyl-peptides) are released in the extracellular environment and circulation and operate as damage-associated molecular pattern (DAMPs), which activate immune responses by binding to pattern recognition receptors (PRRs) of innate immune cells [277]. Intracellularly and upon mitochondrial damage, mtDNA and ROS released in the cytosol can engage either the NLRP3 inflammasome or AIM2 inflammasome, which will mediate the proteolytic maturation and secretion of potent pro-inflammatory cytokines (such as IL-1β and IL-18) through the activation of caspase-1, thus eliciting pro-inflammatory responses [278]. Cytosolic mtDNA can also be sensed by cGAS, which upon stimulation will engage the ER-resident protein STING into activating the Type-I interferon transcriptional program [279]. 

Dysfunctional mitochondria not only provide potent DAMPs, but also sense and decode danger signals by operating as a signaling platform for the recruitment and modulation of the molecular machinery detecting incoming cellular damage. Typically, during viral infection, mitochondria stimulate antiviral signaling through the recruitment of RIG-I and MDA5, two members of the NOD-like receptor family of PRRs, via the mitochondria-associated adaptor protein MAVS (for an extensive review on this subject, please see [280]. Interestingly, MAVS interacts with essential mediators of mitochondrial dynamics (MFN1 and TBC1D15) and this interaction would modulate STING and NLPR3 inflammasome signaling as well [281]. Moreover, NLRP3 and STING are particularly enriched at the ER-mitochondria contact sites (MAMs; [282,283]), ER microdomains that have also been shown to be critical for PINK1/Parkin-mediated mitophagy initiation [284,285,286]. These observations together suggest that mitophagy is intimately linked to mitochondrial sensing and decoding of intracellular danger signaling. In fact, in PINK1- or Parkin-deficient mice the increased oxidative stress caused by mitochondrial iron accumulation in pancreatic cells resulted in AIM2 inflammasome activation, ultimately leading to the release of HMGB1, a non-histone nuclear protein that once released in the extracellular environment upon cell death operates as a DAMP [127]. In this particular case, HMGB1 was released in its oxidized form, which can bind the AGER receptor on innate immune cells and trigger PD-L1 expression, hereby exerting an immunosuppressive role [127]. The concept of PINK1/Parkin- mediated mitophagy as an inflammasome dampener is supported by recent compelling evidence regarding the cargo receptor p62. Upon activation of NF-κB signaling by the NLRP3 inflammasome, parallel mitochondrial damage is induced to trigger Parkin/p62-mediated mitophagy in order to prevent excessive IL-1β-dependent inflammation [287,288]. The activation of the NLPR3 inflammasome signaling is critically dependent on the SESN2-mediated recruitment of p62 to the ubiquitylated mitochondrial membrane [289]. 

Thus, mitophagy operates as a self-limiting system to protect cells from exacerbated inflammation by removing bona fide activators of the NLRP3 inflammasome (ROS, iron and mtDNA). Given the pleiotropic effects in inflammation, immunosurveillance and therapy responses of IL-1β and IL-18, the activation of the inflammasome in cancer may have contextual pro-tumorigenic or anti-tumorigenic roles (for a review on the topic, see [290]). 

On the other side, mitophagy has also been shown to modulate the adaptive immune response in terms of dendritic cell-T cell synapse [291], activation of both CD8+ T cell [244] and memory NK cells [292], suggesting that additional fine-tuning of the immune responses by mitophagy may occur through mechanisms that do not involve the inflammasome. However, more studies are required to fully appreciate the role of mitophagy in cancer inflammation and anti-tumor immunity.

## 4. Mitophagy and Anti-Cancer Therapies

Historically, mitophagy has been considered a protective mechanism used by the cancer cells against the onset of mitochondrial apoptosis, the main cell death pathway driven by the metabolic stress of the TME in cancer cells. Considering that several classes of anti-cancer drugs or treatments either directly or indirectly cause mitochondrial dysfunction, ROS production and cytochrome c release launching caspase activation, targeting these dysfunctional mitochondria by mitophagy could hamper the initiation of the apoptotic cascade. Therefore, the induction of mitophagy by anti-cancer therapies may modulate their cytotoxic ability and contribute to therapy resistance. 

This assumption is supported by several examples where the genetic inhibition of mitophagy pathways sensitize cancer cells towards cell death induced by anticancer treatments [293,294]. In fact, autophagy (and mitophagy by extension) is highly sensitive to microtubule-dynamics modulators since microtubules are essential for the autophagosome fusion with the lysosome [295,296]. In particular, downregulation of PINK1/Parkin- or Rab9a-mediated mitophagy contributes to radiosensitizing cancer cells [297,298], and genetic downregulation of key mitophagy receptors such as PINK1, FUNDC1 or AMBRA1 also chemosensitizes cancer cells [170,299,300], supporting the pro-survival role of mitophagy in cancer cells in response to chemo/radiotherapy.

Considering this pro-survival role of mitophagy after anti-cancer treatment, it could be expected that therapy-resistant cells would display higher mitophagy levels. Indeed, enhanced mitophagy contributes to cisplatin and etoposide resistance in cancer cells [301] and mitophagy impairment resensitizes drug-resistant cancer cells [302,303]. 

However, different mitophagic pathways may be engaged to counteract therapy-induced mitochondrial damage and contribute to the therapy resistance of cancer cells. For example, in docetaxel-treated prostate cancer cells, trehalose induces ULK1-independent mitophagy and reduces their sensitivity to their treatment [304]. Moreover, Opa-interacting protein 5 (OIP5) overexpression prevented docetaxel-induced cell death in gastric cancer cells by activating MFN2/PINK1-mediated mitophagy [305]. Instead, NIX-mediated mitophagy drives doxorubicin-resistance [306], whereas in the case of cisplatin/etoposide, the E3 ubiquitin-ligase ARIH1 -rather than Parkin- mediates PINK1-induced mitophagy of damaged mitochondria in response to cisplatin/etoposide [301]. Together, these studies suggest that the expression of certain receptors or mediators of mitophagy in specific cancer subtypes is a decisive factor in eliciting therapy resistance by mitochondrial clearance. However, although high levels of mitophagy have been correlated with therapeutic resistance, ULK1-driven mitophagy activation has also been shown to have an antiproliferative effect in therapy-resistant colon cancer cell lines [307], implying that mitophagy could play a dual role in therapy resistance.

Mitochondrial clearance (or alternative pathways of mitochondrial removal) could also be involved in modulating cancer cell non-autonomous response to therapy-mediated cell death. Mitochondrial transfer from surrounding cells via microtubules could also contribute to cancer cell therapy resistance, since mitochondria from endothelial cells or stromal cells have been reported to increase the chemoresistance of breast cancer cells [261] or leukemic cells [257], respectively. 

Although the involvement of mitophagy has not been formally proven, blocking the release of the mitochondrial DAMP N-formyl peptides during immunogenic cancer cell death elicited in response to anthracyclines and reduces anti-tumor immunity [308], suggesting a potential regulatory role for mitochondrial clearance, an aspect that deserves further investigation.

When talking about therapy resistance, CSCs deserve special mention. CSCs are a subpopulation within cancer cells which is particularly resistant to therapy and which is responsible for the tumor repopulation after treatment (for a recent review on the topic, see [309]). Interestingly, the ability of CSCs to resist therapy seems to correlate to their mitophagic capacity, since higher mitophagic levels were detected in cisplatin-resistant oral squamous cell carcinomas’ CSCs [310], as well as in oxaliplatin-resistant or doxorubicin-resistant human colorectal CSCs [306,311]. In fact, blockade of mitophagy using a nanomedicine (188Re-Liposome) in chemotherapy-resistant ovarian cancer (stem) cells restored their sensitivity to the treatment not only in xenograft mice but also in two proof-of-concept therapy-resistant cancer patients [312]. Regarding the role of mitophagy in CSC-chemoresistance, it has been pointed out that higher mitophagic levels contribute to a higher OXPHOS state within the CSC population to facilitate its proliferation [313]; although in the case of therapy-resistance, it is more likely that mitophagy, by removing the damaged mitochondria, tightly controls ROS production and mitochondrial transmembrane potential, eventually reducing apoptotic cell death [314]. 

Altogether, preserving mitochondrial fitness by mitophagy provides cancer cells a mechanism to resist anti-cancer therapy (Figure 3); nevertheless, further research is necessary to understand which mitophagy players are instrumental for cancer drug- or radio-resistance in order to develop novel therapeutic strategies aiming to proficiently harness mitophagy in anti-cancer therapy.

### Mitophagy and Cancer-Related Side Effects

Lastly, mitophagy is also relevant for off-target toxicities derived from anti-cancer therapies as well as from cancer itself. In terms of chemotherapeutics, one of the major side-effects related to doxorubicin treatment is its cardiotoxicity caused by enhanced Parkin-mediated or BNIP3-mediated mitophagy in cardiomyocytes [315,316,317]. 

Nonetheless, the most life-threatening side effect of cancer is cancer-associated cachexia: it occurs in more than 80% of the late-stage patients, it is directly associated to 20% of cancer-related deaths [318], and it constitutes a major prognostic factor independently of the cancer type [319]. Cachexia is multifactorial syndrome characterized by muscular mass and adipose tissue loss together with anorexia and weakness [320] and in cancer not only develops particularly fast [321] but it can also be induced by the chemotherapeutic treatment [322]. Cachectic muscle waste and loss have been associated to hypercatabolic breakdown of the muscle by autophagy: LC3B-II, ATG5 and Beclin1 have been shown to accumulate in skeletal muscle tissue from cachectic cancer patients [323,324,325] and in cancer mouse models [326,327]. Recently, it has been pointed out that excessive mitophagy in skeletal muscle cells would promote cachexia development. Alterations in BNIP3 or PINK1 [325] transcript levels are found in cachectic muscle of cancer patients [323,325,328] as well as in mouse models [329,330], but whether and how they regulate mitophagy-induced cachexia remains to be further studied.

In different cancer mouse models (and independently of chemotherapeutic treatment), cachectic muscle displayed alterations in mitochondrial homeostasis [331] (such as mitochondrial uncoupling, aberrant mitochondrial expression of CL [332,333], as well as giant mitochondria unable to fuse proficiently [334]), which would be then targeted for mitophagic degradation. Of note, UCP3, a mitochondrial protein that has been shown to accumulate in cancer-induced cachexia [334], is under the transcriptional control of Tumor Necrosis Factor alpha (TNFα) [335] and certain pro-inflammatory cytokines seem to be highly pro-autophagic/pro-mitophagic in cachectic environments. For example, TNFα and IL-6 trans-signaling induced by the cancer cells accelerates autophagy/mitophagy in skeletal muscle, hereby promoting cachexia [326,336]. Intriguingly, AMBRA1 has been associated with muscular atrophy [337], and has been recently shown to modulate the IL6-STAT3 axis [338], although no direct link to cancer-associated cachexia has been drawn yet.

In contrast to the putative detrimental effect of mitophagy in cancer patients, there are few cases where mitophagy induction in healthy tissue would be beneficial instead: renal failure is a side effect of cisplatin treatment and higher DRP1 levels and mitophagic levels in the renal tubular cells protect them from such a damage [339]. In fact, rapamycin administration reduces cisplatin-mediated nephrotoxicity in C57Bl6 mice by stimulating PINK1/Parkin-mediated mitophagy in tubular renal cells [340]. This suggests that rapamycin, possibly by enhancing mitophagy, could diminish chemotherapy-driven tissue damage. 

Altogether, concomitant mitophagy modulation during chemotherapeutic treatment might contribute not only to overcoming cancer-cell resistance, but also to diminishing side toxicities derived from the treatment or the cancer itself. However, more mechanistic studies are necessary to understand the extent of the relationship between cancer-associated mitophagy and healthy tissue as well as the mitophagic pathways involved.

## 5. Conclusions and Perspectives

In the past years, the world of selective autophagy has grown tremendously, as has our understanding of the molecular underpinnings and physio-pathological implications of selective autophagy pathways. Among various selective degradation pathways operating in mammalian cells, mitophagy is emerging as a crucial determinant of cancer cell plasticity and interface with the TME. Reflecting the dynamic and plastic role of autophagy in carcinogenesis, mitochondrial clearance in cancer appears to operate as a mechanism recruited on-demand by the developing cancer cells to modulate key malignant features during cancer initiation and development. Growing evidence shows that mitophagy pathways act as key regulators of cancer cell mitochondrial mass, dynamics, redox homeostasis, bioenergetics, oncogene-driven metabolic reprogramming and cell death signals. This is perhaps not surprising, considering how mitochondrial biology and metabolic plasticity are central to cancer growth and response to anticancer therapies. The emerging view is that mitophagy represents a flexible mechanism supporting the metabolic adaptation and survival of cancer cells within the harsh TME.

However, from a therapeutic perspective, the redundancy in mitophagy receptors and alternative pathways for mitochondria clearance, which highlights the vital relevance of this process to maintain homeostasis, poses also key challenges when considering new therapeutic avenues harnessing pro-survival mitophagy in cancer treatment. Likewise, our knowledge of additional (and perhaps mitophagy-unrelated) functions of known mitophagy modulators and their link with oncogenic signals must be deepened in order to be able to advance more effective cancer treatments.

## Figures and Tables

**Figure 1 cells-08-00493-f001:**
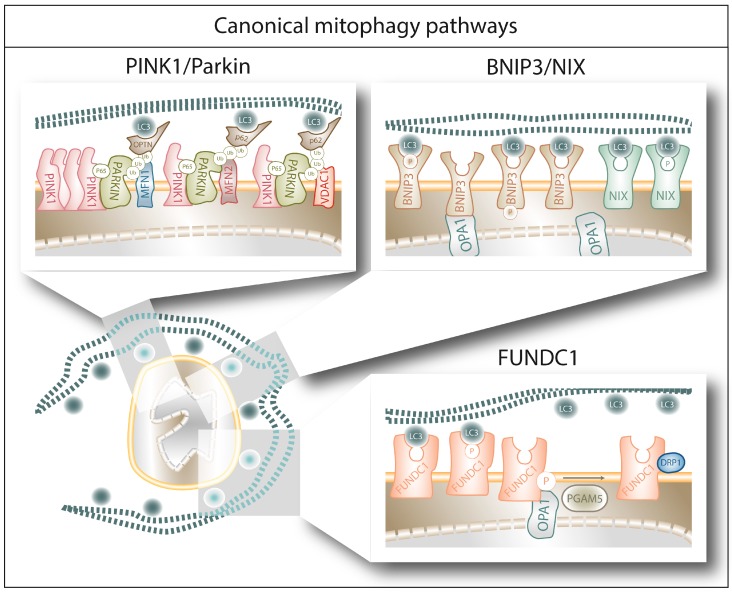
The figure schematically summarizes the main mechanisms and players of canonical mitophagy pathways upon autophagosome recognition of the different receptors in a damaged mitochondrion (see main text for further details). First, PINK1/Parkin mediated ubiquitination of mitochondrial proteins enables the autophagy cargo receptors p62 and OPTN to bridge the mitochondria/autophagosome interaction. Alternatively, BNIP3, NIX and FUNDC1 can directly bind the LC3 molecules decorating the autophagosome, through a mechanism modulated by their phosphorylation status.

**Figure 2 cells-08-00493-f002:**
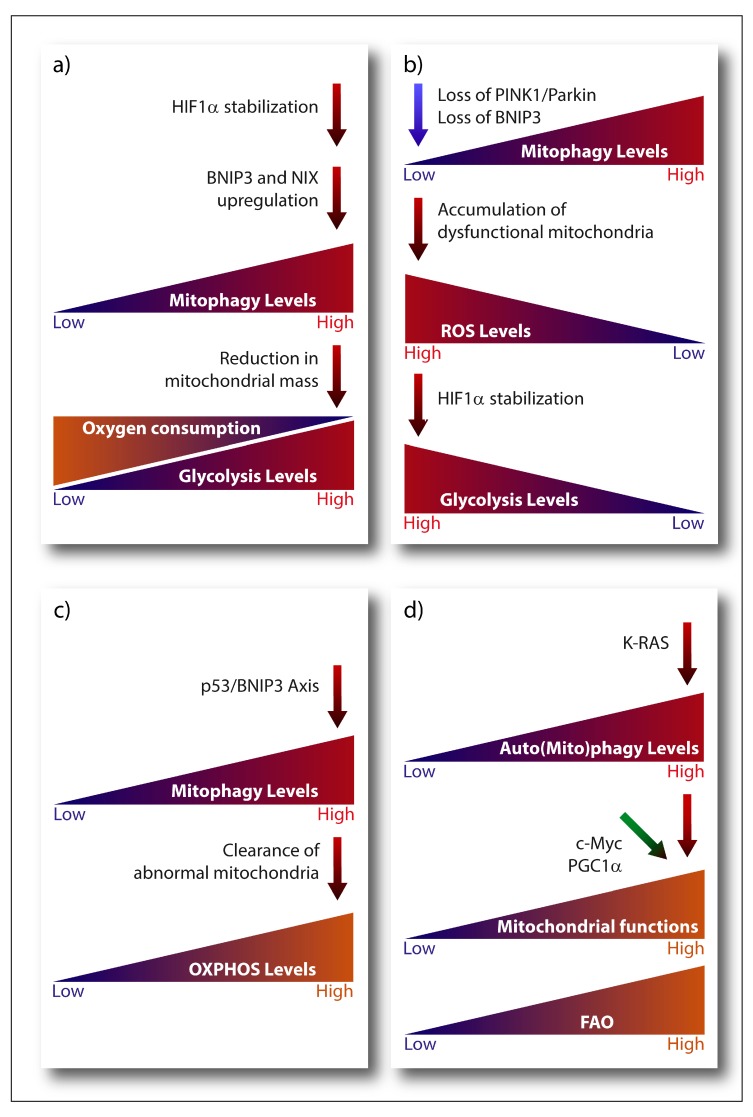
Emerging links connecting oncogene-driven metabolic pathways and key modulators of the mitophagic machinery. (**a**) Upregulation of BNIP3 and NIX expression induced by HIF-1α diminishes mitochondrial mass and O_2_ consumption. (**b**) Loss of BNIP3 or PINK1/Parkin expression reduces mitophagy, leading to the accumulation of damaged mitochondria and ROS, which promotes HIF-1α stabilization and subsequently glycolysis. (**c**) p53 and BNIP3-dependent mitophagy removes abnormal mitochondria, reducing glycolysis and promoting O_2_ consumption. (**d**) RAS driven tumors require elevated autophagy/mitophagy levels to maintain mitochondrial functions and carry out proficient FAO. PGC1α and c-Myc are key regulators of mitochondrial functions to provide cancer cells with metabolic flexibility.

**Figure 3 cells-08-00493-f003:**
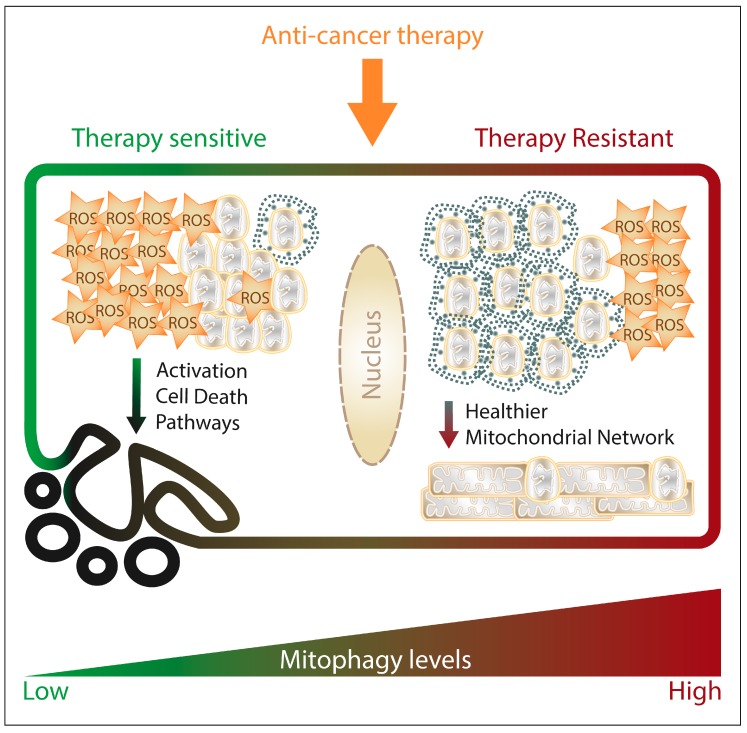
The figure schematically summarizes how mitophagy would modulate the response and resistance of cancer cells to anticancer therapies impinging on mitochondrial functions. Upon anti-cancer treatment, cancer cell mitochondria depolarize and start producing ROS, which if accumulating to lethal levels, trigger apoptosis as major cell death pathway. High mitophagy levels within the treated cancer cell contribute to withstand the damage caused by the treatment and counteract the onset of cell death programs, by maintaining a healthier mitochondrial network (limiting ROS production and accumulation of depolarized mitochondria, preventing cytochrome C release).

**Table 1 cells-08-00493-t001:** Expression levels of mitophagy regulators in samples from cancer patients. The levels of regulation are defined by the correlation of the expression levels with cancer progression and/or poor prognosis.

Protein	Expression Levels in Patients	Cancer Type	Reference(s)
Mitochondrial Dynamics			
MFN 1	Downregulation	Triple Negative Breast Cancer ^1,2^, Hepatocellular Carcinoma ^1,2^	[89,90]
MFN 2	Upregulation	Cutaneous Melanoma ^2^, Gastric Cancer ^2^, Ovarian Cancer ^1^, Prostate Cancer ^2^	[91,92,93,94]
MFN 2	Downregulation	Breast Cancer ^1,2^, Hepatocellular Carcinoma ^1,2^, Non-Small Cell Lung Cancer ^1^	[95,96,97,98,99,100]
Opa1	Upregulation	Lung Cancer ^2,3^, Colorectal Cancer ^1^	[101,102,103]
Opa1	Downregulation	Hepatocellular Carcinoma ^2^	[104]
DRP1	Upregulation	Triple Negative Breast Cancer ^1,2^, Colorectal Cancer^1^, Hepatocellular Carcinoma ^1,2^, Ovarian Cancer ^1,3^	[89,90,101,105]
DRP1	Downregulation	Colorectal Cancer ^2^, Lung Cancer ^2^	[106]
pDRP1 (Ser616)	Upregulation	Colorectal Cancer ^2^, Melanoma ^2^	[107,108]
pDRP1 (Ser637)	Upregulation	Hepatocellular Carcinoma ^2^	[109]
Mff	Upregulation	Hepatocellular Carcinoma ^1,2^	[110]
Mff	Downregulation	Tongue Squamous Cell Carcinoma ^2^	[111]
FIS1	Upregulation	Acute Myeloid Leukemia ^1,2^, Oral Melanoma ^2^, Prostate Cancer ^1^	[93,112,113,114]
FIS1	Downregulation	Tongue Squamous Cell Carcinoma ^2^	[115]
**Canonical Mitophagy Pathways**			
PINK1	Upregulation	Lung Cancer ^2^, Esophageal Squamous Cell Carcinoma ^2^	[116,117]
PINK1	Downregulation	Ovarian cancer ^1^	[118]
Parkin	Downregulation	Acute Lymphoblastic Leukemia ^4^, Colorectal Cancer ^1,2,4^, Clear Cell Renal Cell Carcinoma ^1,2^, Melanoma ^1,3^, Oropharyngeal Squamous Cell Carcinoma ^1^, Ovarian Cancer ^1,3^, Pancreatic Cancer ^1,2,3^	[119,120,121,122,123,124,125,126,127]
BNIP3	Upregulation	Adenoid Cystic Carcinoma ^2^, Ampullary Carcinoma ^2^, Breast Cancer ^1^, Cervical Cancer ^1,2^, Cholangiocarcinoma ^2^, Colorectal Cancer ^2^, Ependydoma ^1^, Glioblastoma ^2^, Lung Cancer ^1,2^, Melanoma ^2^, Ovarian Cancer ^1,2^, Renal Carcinoma ^1,2^, Uterine-Cervical Squamous Cell Carcinoma ^1^	[128,129,130,131,132,133,134,135,136,137,138,139,140,141,142,143]
BNIP3	Downregulation	Bladder Cancer ^4^, Breast Cancer ^1,2^, Colorectal Cancer ^2,4^, Esophageal Cancer ^4^, Gastric Carcinoma ^4^, Laryngeal Squamous Cell Carcinoma ^2^, Lung Cancer ^4^, Multiple Myeloma ^1,4^, Pancreatic Cancer ^1,2^,	[142,144,145,146,147,148,149,150,151,152,153,154,155,156,157,158,159,160,161,162,163]
NIX	Upregulation	Breast Cancer ^1^, Glioma ^1,2^	[164,165]
NIX	Downregulation	Acute Myeloid Leukemia ^1^, Prostate Cancer ^3^	[166,167]
FUNDC1	Upregulation	Breast Cancer ^1,2^, Cervical Cancer ^2^, Laryngeal Cancer ^2^	[168,169,170]
PGAM5	Upregulation	Hepatocellular Carcinoma ^2^, Non-Small Cell Lung Cancer ^2^	[171,172]
**Non-Canonical Mitophagy Pathways**			
CL	Upregulation	Prostate Cancer	[173,174]
CL	Downregulation	Hepatocellular Carcinoma	[175]
PHB2	Upregulation	Breast Cancer ^1,2^, Colorectal Cancer ^2^, Esophageal Squamous Cell carcinoma ^1,2^, Leukemia ^2^, Lymphoma ^2^	[167,176,177,178,179,180]
C18-Ceramide	Downregulation	Glioblastoma, Glioma, Head and Neck Squamous Cell Carcinoma	[181,182,183]
AMBRA1	Upregulation	Cholangiocarcinoma ^2^, Gastric Adenocarcinoma ^2^, Pancreatic Ductal Adenocarcinoma ^2^, Prostate Cancer ^1,2^	[184,185,186,187]
HUWE1	Upregulation	Lung Cancer ^1,2^, Multiple Myeloma ^1^	[188,189]
HUWE1	Downregulation	Breast Cancer ^1^, Hepatocellular Carcinoma ^1^, Osteosarcoma ^1^	[190,191,192]
BCL2L13 (BCL-RAMBO)	Upregulation	Leukemia ^1^	[193,194,195]
BCL2L13 (BCL-RAMBO)	Downregulation	Breast Cancer ^1^, Locally Advanced Rectal Cancer ^1^	[139,196]
RAB7	Upregulation	Oral Squamous Cell Carcinoma ^2^, Prostate Cancer ^1^,	[197,198]

^1^ mRNA expression levels. ^2^ Protein expression levels. ^3^ Copy Number Variation levels. ^4^ Promoter methylation levels.

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
