# Peer review of "Mitophagy in Cancer: A Tale of Adaptation"

_cells, 2019, doi:10.3390/cells8050493_

Reviewer 1 Report

Overall, it is an extensive review which provides a detailed and well-explained overview of all the possible aspects of the mitophagy role in cancer. The authors have discussed various signalling pathways involved in the mitophagy process. They have also emphasized the critical aspects of the cellular crosstalk of mitophagy and the TME, as well as the importance of mitophagy in relation to anti-cancer therapy. The manuscript is well written and proposes detailed illustrations. However, there are some points that should be clarified before acceptance, as follows:

1)     Figures are never recalled in the main text.

2)     Some of the literature used is reviews. The authors are highly recommended to also cite original papers.

3)     The section concerning the mitophagy pathways might be further improved by highlighting the differences among them.

4)     Line 30: change “in charge of […] of the cells” with “deputed at cell energy supply”.

5)     Line 33, 298, 299: rewrite the sentences.

6)     BOX1: 

-       the mechanistic of mitochondrial fusion should be discussed more extensively, at least at the same extent of the fission.

-       The authors should also include the effects of OPA1 cleavage (L-OPA, S-OPA) on mitochondrial dynamics.

7)    Line 117: the authors should also include and specify the mechanistic preventing PINK1 import and cleavage upon ΔΨm loss.

8)    Lines 173, 539: revise citation formatting.

9)    Line 179: the statement should be supported by literature citation.

10) The title of paragraph 3.1.1. should be revised since the Warburg effect is a very wide phenomenon, while the authors focus on few and limited aspects related to it (HIF-1a and ROS, mostly).

11) A more extensive citation of the literature related to the role of p53 on mitophagy should be included.

12) The authors should also include other ROS-mediated pathways (e.g. the MAPK pathway) involved in mitochondrial clearance.

13) Please note that PGC1a expression can be tightly connected to lipid metabolism. Therefore, it is suggested to (at least) discuss/speculate about possible correlations between PGC1a, lipid metabolism/homeostasis and mitophagy.

14) The implication(s) of OXPHOS-mediated ROS production with mitophagy should be discussed.

Author Response

Overall, it is an extensive review which provides a detailed and well-explained overview of all the possible aspects of the mitophagy role in cancer. The authors have discussed various signalling pathways involved in the mitophagy process. They have also emphasized the critical aspects of the cellular crosstalk of mitophagy and the TME, as well as the importance of mitophagy in relation to anti-cancer therapy. The manuscript is well written and proposes detailed illustrations.

We would like to thank Reviewer #1 for his/her constructive and positive comments about our review.

However, there are some points that should be clarified before acceptance, as follows:

1)     Figures are never recalled in the main text.

Following Reviewer #1’s kind suggestion, the figures were referenced throughout the main text.

2)    Some of the literature used is reviews. The authors are highly recommended to also cite original papers.

We appreciate the Reviewer #1’s remark. Considering that our review covers the role of mitophagy in several and complex processes and in order to not overload the review and losing focus on the main topic, we decided to provide the reader other recent and more exhaustive reviews that cover each particular topic in more depth and that could help the reader. However, in response to Reviewer #1’s comment we have revised the literature again to make sure that primary sources were cited as much as possible.

3)    The section concerning the mitophagy pathways might be further improved by highlighting the differences among them.

We value Reviewer #1’s suggestion. Although we appreciate that including a paragraph on the differences among the canonical forms of mitophagy (since for the non-canonical still need further characterization), throughout the review we have noticed –especially in a cancer context- that these differences are mainly context and cancer-type specific. Hence, we thought that they would be addressed better in the specific sections made below, rather than drawing too general conclusions in the part describing the mitophagy pathways. 

4)     Line 30: change “in charge of […] of the cells” with “deputed at cell energy supply”.

Following Reviewer #1’s kind suggestion, the sentence was rephrased accordingly (page 1, line 30).

5)     Line 33, 298, 299: rewrite the sentences.

Following Reviewer #1’s kind suggestion, the sentences were rewritten (now lines 33, 308 and 309).

6)     BOX1:

·         the mechanistic of mitochondrial fusion should be discussed more extensively, at least at the same extent of the fission.

·         The authors should also include the effects of OPA1 cleavage (L-OPA, S-OPA) on mitochondrial dynamics.

We appreciate the Reviewer #1’s remark. In terms of mitophagy and within the context of mitochondrial dynamics, fission (and by extension the lack of fusion) is considered more relevant, hence the deeper elaboration on fission than fusion. Nevertheless, we have slightly expanded the fusion section (page 3, lines 85-93) including Opa1 cleavage as well, as suggested by the Reviewer.

7)    Line 117: the authors should also include and specify the mechanistic preventing PINK1 import and cleavage upon ΔΨm loss.

Following the Reviewer #1’s kind suggestion, a couple of sentences were added (page 4, lines 124-127) on the accumulation of PINK1 in the outer mitochondrial membrane in terms of import and cleavage upon membrane depolarization.

8)    Lines 173, 539: revise citation formatting.

Following the Reviewer #1’s kind suggestion, the format of the citations was corrected and the rest of the citations revised.

9)    Line 179: the statement should be supported by literature citation.

We consider the Reviewer #1’s remark. We meant the sentence as speculative, since there is no literature yet (at least that we could find) that could support that statement. Moreover, there was a typo in the sentence (now lines 188-189) which was corrected and we hope that now it is clear that it is speculation.

10) The title of paragraph 3.1.1. should be revised since the Warburg effect is a very wide phenomenon, while the authors focus on few and limited aspects related to it (HIF-1a and ROS, mostly).

We agree with Reviewer #1 that the Warburg effect is a wide and complex phenomenon; therefore, we made a subtle change in the title of paragraph 3.1.1.

11) A more extensive citation of the literature related to the role of p53 on mitophagy should be included.

We appreciate the Reviewer #1’s remark. Although there are several papers that describe the role of p53 in mitophagy as well as its correlation with some mitophagy players, we decided to refer only to those papers that explicitly describe a connection between p53 and mitophagy within the context of cancer metabolism (page 11, lines 299-305).

12) The authors should also include other ROS-mediated pathways (e.g. the MAPK pathway) involved in mitochondrial clearance.

We thank Reviewer #1 for calling our attention to this point. We have included a couple of sentences (page 12, lines 325-327) to highlight the clearance of damaged mitochondria induced by the MAPK pathway in response to ROS-induced stress.

13) Please note that PGC1a expression can be tightly connected to lipid metabolism. Therefore, it is suggested to (at least) discuss/speculate about possible correlations between PGC1a, lipid metabolism/homeostasis and mitophagy.

This is an interesting point. Following the Reviewer #1’s kind suggestion we have discussed it in page 12, lines 349-354.

14) The implication(s) of OXPHOS-mediated ROS production with mitophagy should be discussed.

Following the Reviewer #1’s kind suggestion, we addressed the topic in page 12, lines 328-329.

Reviewer 2 Report

This is a great review explaining the role of mitophagy in cancer and its potential application for therapy. It is well-structured, starting from the Molecular mechanisms of mitophagy and the pathways to finally explain its implication in cancer. I particularly liked the table 1 with the expression levels of mitophagy regulators from cancer patients. I think it is really useful for future research in this field. 

Before the full acceptance of the manuscript, I think it will improve if the authors consider to include the following:

- abbreviation list. There are a lot of proteins,... I suggest to include an abbreviation list at the end.

- will it be possible to expand the section Mitophagy and anti-cancer therapies? Or at least including a table? I think it will be really useful for future studies.

Congratulations for your manuscript! It is a great contribution.

Author Response

This is a great review explaining the role of mitophagy in cancer and its potential application for therapy. It is well-structured, starting from the Molecular mechanisms of mitophagy and the pathways to finally explain its implication in cancer. I particularly liked the table 1 with the expression levels of mitophagy regulators from cancer patients. I think it is really useful for future research in this field.

We would like to thank Reviewer #1 for his/her constructive and positive comments about our review.

Before the full acceptance of the manuscript, I think it will improve if the authors consider to include the following:

- abbreviation list. There are a lot of proteins,... I suggest to include an abbreviation list at the end.

Following the Reviewer’s #2 kind suggestion, we added an abbreviation list at the end of the manuscript (page 27).

- will it be possible to expand the section Mitophagy and anti-cancer therapies? Or at least including a table? I think it will be really useful for future studies.

We appreciate the Reviewer #1’s suggestion. Initially, the section covering anti-cancer therapies was much longer since our criteria would just consider the involvement of pro-mitophagic proteins. However, we soon noticed that some of the studies were describing cell-death related effects and not mitophagy per se (especially when it came to BNIP3 and NIX, two mitophagy receptors that also belong to the BCL2 family and have contextual pro-cell death functions). Therefore, we went for criteria that are more stringent and select only those studies that explicitly mentioned mitophagy, hereby significantly shortening the section as it is.

Congratulations for your manuscript! It is a great contribution.